# Dietary Exposure to Flame Retardant Tris (2-Butoxyethyl) Phosphate Altered Neurobehavior and Neuroinflammatory Responses in a Mouse Model of Allergic Asthma

**DOI:** 10.3390/ijms23020655

**Published:** 2022-01-07

**Authors:** Tin-Tin Win-Shwe, Rie Yanagisawa, Thet-Thet Lwin, Fumitaka Kawakami, Eiko Koike, Hirohisa Takano

**Affiliations:** 1Health and Environmental Risk Division, National Institute for Environmental Studies, Tsukuba 305-8506, Japan; yanagisawa.rie@nies.go.jp (R.Y.); ekoike@nies.go.jp (E.K.); 2School of Allied Health Sciences, Kitasato University, Sagamihara 252-0373, Japan; ttlwin@kitasato-u.ac.jp (T.-T.L.); kawakami@kitasato-u.ac.jp (F.K.); 3Graduate School of Global Environmental Studies, Kyoto University, Kyoto 615-8530, Japan; takano.hirohisa.4x@kyoto-u.ac.jp

**Keywords:** tris (2-butoxyethyl) phosphate (TBEP), novel object recognition test, inflammatory biomarkers, hippocampus, allergic asthmatic mouse

## Abstract

Tris (2-butoxyethyl) phosphate (TBEP) is an organophosphate flame retardant and used as a plasticizer in various household products such as plastics, floor polish, varnish, textiles, furniture, and electronic equipment. However, little is known about the effects of TBEP on the brain and behavior. We aimed to examine the effects of dietary exposure of TBEP on memory functions, their-related genes, and inflammatory molecular markers in the brain of allergic asthmatic mouse models. C3H/HeJSlc male mice were given diet containing TBEP (0.02 (TBEP-L), 0.2 (TBEP-M), or 2 (TBEP-H) μg/kg/day) and ovalbumin (OVA) intratracheally every other week from 5 to 11 weeks old. A novel object recognition test was conducted in each mouse at 11 weeks old. The hippocampi were collected to detect neurological, glia, and immunological molecular markers using the real-time RT-PCR method and immunohistochemical analyses. Mast cells and microglia were examined by toluidine blue staining and ionized calcium-binding adapter molecule *(Iba)-1* immunoreactivity, respectively. Impaired discrimination ability was observed in TBEP-H-exposed mice with or without allergen. The mRNA expression levels of *N*-methyl-*D* aspartate receptor subunits *Nr1* and *Nr2b*, inflammatory molecular markers tumor necrosis factor-α oxidative stress marker heme oxygenase 1, microglia marker *Iba1,* and astrocyte marker glial fibrillary acidic protein were significantly increased in TBEP-H-exposed mice with or without allergen. Microglia and mast cells activation were remarkable in TBEP-H-exposed allergic asthmatic mice. Our results indicate that chronic exposure to TBEP with or without allergen impaired object recognition ability accompanied with alteration of molecular expression of neuronal and glial markers and inflammatory markers in the hippocampus of mice. Neuron-glia-mast cells interaction may play a role in TBEP-induced neurobehavioral toxicity.

## 1. Introduction

Tris (2-butoxyethyl) phosphate (TBEP) is one of the organophosphate flame retardants (OPFRs). Flame retardants (FRs) are extensively used in construction materials, textiles, and electrical appliances around the world. Some of FRs, such as polybrominated diphenyl ethers and hexabromocyclododecane, have been banned for toxicity, and OPFRs are used to replace toxic FRs. Therefore, it is necessary to examine the biological effect of exposure to TBEP, which is a commonly used OPER. Recent studies have shown that OPFRs have neurotoxic effects and potential carcinogenic effects in different animal studies [1,2,3,4]. OPFRs are released into the environment from consumer products, and humans can be exposed OPFRs easily via ingestion, inhalation, and dermal contact. TBEP was detected in indoor air, house dust, soil, and water. It has been reported that daily exposure to OPFRs via indoor dust [5] and diet and diet-associated intake is higher [6]. Thus, it was suggested that common routes of entry of TBEP to the body are inhalation of indoor dust or ingestion of contaminated food.

There are rising numbers of OPFRs-induced environmental health risks, including carcinogenicity, embryo toxicity, teratogenicity, immunotoxicity, neurotoxicity, and endocrine-disrupting activity in humans and animals [7]. A single oral dose greater than LD50 or 5000 mg/kg TBEP decreased approximately 70% of plasma butyrylcholinesterase activity and approximately 45% of acetylcholinesterase activity in the adult hen brain [8]. Early life stage exposure to TBEP decreased free-swimming speed significantly and caused downregulation of the transcriptions of nervous system genes in zebrafish larvae suggesting that TBEP disrupts the developing nervous system [9]. In addition, TBEP exposure in early life stages showed hatchability reduction, delay time to hatching, increased the occurrence of malformations, body length reduction, slow heart rate, hypoactivity, and changes of acetylcholinesterase activity and transcriptional genes related to the nervous system in Japanese medaka [10]. Recently, our research group has reported that dietary exposure of low dose TBEP exacerbates pulmonary inflammation, estrogen receptor α expression in the lungs, and cellular responses in the mediastinal lymph nodes and bone marrow in allergic asthmatic mouse models [11]. However, the neurotoxic effects of TBEP are still limited, and mechanisms of toxicity are not well known.

Previously, we have described possible brain–lung networks in asthma. Briefly, asthma may induce brain hypoxia and cause inflammatory responses via neuro-immune molecular markers, which induce cognitive impairment via hippocampal memory function-related genes. We have reported that exposure to environmental chemicals enhanced these effects [12,13,14]. A recent study has also demonstrated that asthma-induced brain hypoxia caused the impairment of learning and memory via dysfunction of synaptic tissues and blood vessels, increased levels of hypoxia-inducible factors marker *Hif-1α* and *Hif-2α* [15]. Therefore, in this study, we hypothesized that chemical alone, asthma alone, or co-exposure of chemical and allergen induces brain hypoxia causes memory impairment via modulation of hippocampal synaptic genes and inflammatory mediators released from brain immune and glial cells. The purpose of this study was to detect the effects of dietary exposure of TBEP on memory function, memory function-related genes, and inflammatory molecular markers in the brain of allergic asthmatic mouse models.

## 2. Results

### 2.1. Assessment of General Toxicity

Body and brain weights of the mice were measured at the time of sampling to detect general toxicity effects. No statistically significant difference was observed among the groups of mice exposed to TBEP with or without OVA immunization (Figure 1).

### 2.2. Allergic Asthmatic Mouse Model

To confirm the allergic asthmatic mouse model, we measured the cellular profile of bronchoalveolar (BAL) fluid 48 h after final intratracheal instillation of OVA and serum OVA-specific immunoglobulins (Ig) in the same OVA-immunized mice and described in our previous report [11]. TBEP treatment increased eosinophils and lymphocytes in BAL fluid of OVA-immunized mice compared to vehicle-treated group. Moreover, significantly increased OVA-specific IgE and IgG1 were observed in OVA-treated groups compared with the vehicle group [11]. These results indicate that OVA-treated mice show airway hyperresponsiveness and allergic asthmatic condition.

### 2.3. Effect of TBEP Exposure on Novel Object Recognition Test

A novel object recognition test was performed to detect memory function. No association with absence or presence of OVA, TBEP-H, and OVA + TBEP-H group showed significantly reduced discriminating ability compared to the vehicle-alone group (*p* < 0.05, Figure 2). It was suggested that the vehicle-alone, TBEP-L, TBEP-M groups can recognize a novel object, and OVA-treated groups had a poor ability to discriminate between familial and novel ones in OVA alone (*p* = 0.76 vs. vehicle), TBEP-L (*p* = 0.54 vs. vehicle) and TBEP-H groups. Among them, TBEP-H and OVA + TBEP-H groups showed significantly reduced discriminating ability compared to the vehicle-alone group.

### 2.4. Effect of TBEP Exposure on Memory Function-Related GeneNmda Receptor Subunits in the Hippocampus

We investigated the effect of TBEP exposure on memory function-related genes *Nmda* receptor subtype *Nr1*, *Nr2a* and *Nr2b* in the hippocampus of mice with or without OVA immunization. We found that the expression levels of *Nr1* and *Nr2b* mRNAs were significantly increased in TBEP-H and OVA + TBEP-H exposed group compared with the vehicle-alone group (*p* < 0.05, Figure 3A,B). Regarding *Nr1*, OVA + TBEP-H exposed group showed significantly increased *Nr1* mRNA compared with the OVA-alone group (*p* < 0.05, Figure 3A).

### 2.5. Effect of TBEP Exposure on Inflammatory Molecular Markers in the Hippocampus

To detect the TBEP-induced inflammation in the hippocampus, we have also investigated the inflammatory molecular markers such as interleukin *(Il)-1β*, tumor necrosis factor *(Tnf) α* and heme oxygenase *(Ho)-1* and ionized calcium-binding adapter molecule *(**Iba)1* in the hippocampus of mice with or without OVA immunization. *Il-1**β* mRNA expression was significantly increased in the OVA + TBEP-H group compared with the vehicle group (*p* < 0.05; Figure 4A). The expression levels of *Tnf-α*, *Ho1*, and *Iba1* mRNAs were remarkably increased in mice exposed to TBEP-H with or without OVA compared with the corresponding vehicle-alone and OVA-alone groups (*p* < 0.05; Figure 4B–D).

### 2.6. Effect of TBEP Exposure on Astrocyte Marker Gfap in the Hippocampus

Activated astrocytes express an increased level of glial fibrillary acidic protein (*Gfap*), which is used as a marker protein for astrogliosis. We examined the expression level of *Gfap* and found *Gfap* mRNA was significantly increased in TBEP-M and TBEP-H groups with or without OVA compared with the vehicle-alone or OVA-alone groups (*p* < 0.05, *p* < 0.01; Figure 5).

### 2.7. Effect of TBEP Exposure on Histological Changes in the Hippocampus

We have investigated the effect of TBEP exposure on histological changes in the dentate gyrus of the hippocampus of OVA-immunized allergic asthmatic mice. We did not find any significant changes of morphology between vehicle and OVA-treated groups. Although H&E stain showed no remarkable difference (Figure 6), we further examined the brain immune cells such as microglia and mast cells with specific staining techniques.

### 2.8. Effect of TBEP Exposure on Microglia Marker Iba1 Immunoreactivity in the Hippocampus

The activation of the major immune cells, microglia, in the hippocampus of the vehicle and TBEP-exposed allergic asthmatic mice was examined by using the microglial marker *Iba1*. We found that microglial activation was markedly increased in the hippocampus of the OVA + TBEP-H-exposed group as compared with that in the vehicle group. Representative digital photomicrographs of *Iba1*-immunostained sections taken from the hippocampus of the vehicle and TBEP-exposed groups with OVA immunization are shown in Figure 7. *Iba1*-positive microglia were quantified under a high power field in the DG area of the hippocampus using ImageJ software. *Iba1*-positive microglia were significantly increased in the OVA + TBEP-H-exposed group as compared with that in the vehicle, OVA and OVA + TBEP-L groups.

### 2.9. Effect of TBEP Exposure on Mast Cell Expression in the Hippocampus

The mast cell is one brain immune cell, a small number under basal conditions, and becomes a large number under neuroinflammation. Mast cells were detected in the dentate gyrus of the hippocampus of OVA-immunized allergic asthmatic mice using toluidine blue. We found that mast cells were remarkably increased in the OVA + TBEP-H group compared to vehicle, OVA and OVA + TBEP-L, OVA + TBEP-M groups (Figure 8).

### 2.10. Effect of TBEP Exposure on Fc Epsilon R1a (FcεR1α) Expression in Mast Cell in the Hippocampus

Mast cells express the high-affinity receptor for IgE (*FcεR1α*) on their surface, and these can be activated by IgE and specific antigens to release various mediators. *FcεRIα*-positive mast cells were determined in the dentate gyrus of the hippocampus of OVA-immunized allergic asthmatic mice using the immunostaining method. We found that *FcεRIα*-positive mast cells were remarkably increased in the OVA + TBEP-H group compared to vehicle, OVA and OVA + TBEP-L, OVA + TBEP-M groups (Figure 9). *FcεR1α*-positive mast cells were quantified under a high power field in the DG area of the hippocampus using ImageJ software.

## 3. Discussion

The major findings of the present study were that oral exposure to TBEP (1) impairs novel object recognition ability accompanied with abnormal activation of memory function-related gene *Nmda* receptor subunits, (2) induces upregulation of proinflammatory cytokine *Il-1β*, *Tnf-α* and oxidative stress marker *Ho1*, and (3) activates brain immune markers such as microglia and mast cells in the hippocampus of mice exposed to TBEP-H with OVA immunization. Unfortunately, we did not detect activation of microglia and mast cells in mice exposed to TBEP-only groups. Our findings suggest that oral exposure to TBEP induces cognitive deficit in mice by triggering neuroinflammation and changes in neurological and immunological molecular markers in the brain.

Glutamate is an excitatory amino acid neurotransmitter, and glutamate receptors mainly consist of *Nmda* and α-amino-3-hydroxy-5-methylisoxazole-4-propionic acid (*Ampa*) receptors. The *Nmda* receptor is associated with the induction of synaptic plasticities such as long-term potentiation (LTP) and long-term depression (LTD), which play a major role in learning and memory functions [16]. Tris (2-chloroethyl) phosphate (TCEP) is one of the organophosphorus flame retardants. Six-week-old female SD rats exposed to TCEP (250 mg/kg/d daily by oral gavage for one month) showed impairment of amino acid neurotransmitter metabolism, energy metabolism, and cell membrane integrity inducing neurodegeneration, and neuronal loss in the hippocampal CA1 region and resulted in poor learning and memory functions of rats [17]. Moreover, brominated flame-retardant tetrabromobisphenol A (TBBPA, 25 μM)-induced glutamate release at synaptic regions and induced depolarization mediated by *Nmda* receptors and *Ampa* receptors in both rat cerebellar granule cell-line and primary cultures [18]. *Nr2a*, *Nr2b* and some *Nr1* are the central *Nmda* receptor subunits expressed in the brain of rodents. The hippocampus is the essential area during the brain developmental process accounting for cognition, learning, and memory. In the present study, we investigated the effect of TBEP exposure on *Nmda* receptor subtype *Nr1*, *Nr2a* and *Nr2b* in the hippocampus of mice with or without OVA immunization. We found that the expression levels of *Nr1* and *Nr2b* mRNAs were significantly increased in TBEP-H and OVA + TBEP-H exposed group compared with the vehicle-alone group (Figure 3). OVA + TBEP-H exposed group showed significantly increased *NR1* mRNA compared with the OVA-alone group. The increased mRNA expression levels of *Nmda* receptor subunits may be due to increased neurotoxic glutamatergic activity [19] and then may cause neuronal damage, which in turn may trigger impairment of novel object recognition ability in TBEP-H exposed mice with or without OVA immunization.

Tris (1,3-dichloro-2-propyl) phosphate (TDCPP) is one of the most common OPFRs. Oral TDCPP exposure in C57BL/6 pups (0, 5, or 50 mg/kg/day) during the neonatal period upregulates the mRNA levels of proinflammatory cytokines *Il-1β*, *Tnf-α*, C-C motif chemokine ligand 2 (*Ccl2*) and activates microglia in hippocampi after one day and 28 days of treatment [20]. In the present study, *Il-1β* mRNA expression was significantly increased in the OVA + TBEP-H group compared with the vehicle group, and *Tnf-α*, *Ho1,* and *Iba1* mRNAs were remarkably increased in mice exposed to TBEP-H with or without OVA compared with the corresponding vehicle-alone and OVA-alone groups (Figure 4). In vitro study has demonstrated that 2,2′,4,4′-tetrabromodiphenyl ether (BDE-47), brominated flame retardants, induces oxidative stress and apoptotic cell death in mouse cerebellar granule neurons [21]. Furthermore, in vivo study, male *Gclm* +/+ and *Gclm* −/− mice treated with oral BDE-47 (10 mg/kg) on PND 10 induces oxidative stress via MDA, 8-isoprostane, and reactive protein carbonyls in the cerebellum [21].

Microglia and mast cells are natural immune cells in the brain and play a key role in neuroinflammation. Microglia are major immune effector cells in the brain and, such as mast cells, a major player in neuroinflammation. Microglia detect the changes in surrounding and provide immunosurveillance activity. The functions of the microglia are maintaining neuronal synapses, identifying pathogens, and removing cellular debris [22]. In this study, prominent microglia activation and significantly increased microglia marker *Iba1* expressions were detected in the OVA + TBEP-H group (Figure 7).

Mast cells serve as sensors for environmental changes and first responders to communicate with other cellular mediators that are involved in immune responses and enhance the neuroinflammatory process [23]. The major function of mast cells is degranulation, and activated mast cells release mediators such as histamine, tryptase, chymase, interleukins, *Tnf-α*, serotonin, prostaglandins, chemokines, and growth factors into the interstitium [24,25]. In the present study, by using mast cells surface marker *FcεR1α*, we found mast cell activation is prominent in the OVA + TBEP-H group (Figure 8). In addition, mast cell degranulation was reported to cause a cognitive deficit in rats [26]. Di (2-ethylhexyl) phthalate (DEHP) is a hazardous chemical that is used as a plasticizer in plastic products. Previously, we have shown that intratracheal administration of DEHP during the juvenile period induced neuroinflammation by modulating the neuroimmune biomarkers such as *Il-1β* and *Tnf-α*, *Ccl3*, *Nfκb*, *Ho1*, nerve growth factor, and the microglia marker *Iba1* in the hypothalami of allergic asthmatic mouse models [22]. We have also demonstrated that intratracheal instillation of bisphenol A (BPA), which is a raw material of plastic products, may induce increased airway inflammatory response via Th2 responses, lung hormone receptor reduction, mediastinal lymph node, and bone marrow microenvironment alteration [27,28]. With a similar experimental protocol, we found that impairment of novel object recognition ability accompanied with *Nmda* receptor subunit changes in the intratracheally BPA-exposed allergic asthmatic mice [29]. Moreover, recently, we have reported that dietary exposure to BPA may affect higher brain functions by modulating neuroimmune biomarkers in allergic asthmatic mice [14]. Triphenyl phosphate (TPHP) is used as an alternative to brominated flame retardant. Oral exposure to TPHP from PND 10-70 reduced the level of neuronal axon marker (TUBB3), pre- and postsynaptic markers synaptophysin, downregulated the gene expression of neurotransmitter receptors such as *Nmda* receptor (*Grin2b*, *Grin2c*, *Grin2d*, *Grin3a*), serotonin receptor (5-hydroxytryptamine receptor 1A (*Htr1α*)), alpha-adrenergic receptor (*Adra1α*), and inhibited synaptic proteins (*Stx1a*, *Syt1*) for calcium signal in hippocampal neurons and resulting in learning and memory impairment in mice [30].

Astrocytes are another major glial cell in the brain and play a role in brain homeostasis by inducing the release of several neurotrophic factors. Like microglia, astrocytes can react to various environmental stress and leading to astrogliosis [31]. Normally, activated astrocytes secrete different neurotrophic factors for neuronal survival; however, abnormally activated astrocytes secrete various neurotoxic substances and induce an inflammatory response that triggers neuronal death [32]. Abnormally or severely activated astrocytes express an increased level of glial fibrillary acidic protein (*Gfap*), which is used as a marker protein for astrogliosis [33]. In this study, *Gfap* mRNA expression was significantly increased in the hippocampus of mice exposed to TBEP-H with or without OVA immunization. It was suggested that TBEP-H exposure may induce microgliosis, astrogliosis, and neuronal impairment simultaneously, and these neuronal and two glia cells interactions may induce neuroinflammation and neurobehavioral impairment.

In our previous studies, not chemical-only exposure, co-exposure to chemical (e.g., DEHP [13], BPA [14,29]) and allergen (OVA) during the juvenile period to adulthood impaired novel object recognition ability, accompanied by alterations in the expression level of memory function-related genes and neuroimmune molecular markers in the brain. In the present study, we hypothesized that co-exposure of TBEP with allergen induced neuroinflammation via neurological and immunological molecular markers, which causes neurobehavior abnormality. However, TBEP-only-treated groups also showed poor novel object recognition ability and alterations in the expression level of memory function-related genes and inflammatory molecular markers expressions. The limitation of this study was the lack of examination of microglia and mast cells activation in TBEP-only-treated groups. We have a plan to examine the effects of chemical-only exposure and chemical and allergen co-exposure on histology and immunoreactivity of brain immune cells using animal models in future studies.

Taken together, similar to the other environmental chemicals, oral TBEP exposure induces neuroinflammation via activated microglia-released cytokines, oxidative stress causing neural damage in the hippocampus, especially in synaptic regions via abnormal activation of *Nmda* receptor subunits, which may be due to increased glutamatergic activity. This neuroinflammation and neuronal damage in the hippocampus may lead to impairment of novel object recognition ability in TBEP-H exposed mice.

## 4. Materials and Methods

### 4.1. Animals

Four-week-old male C3H/HeJSlc mice were purchased from Japan SLC, Inc. (Shizuoka, Japan) and used in this study as described previously [11]. Briefly, five-week-old mice were allotted into eight groups as follows: (1) vehicle, (2) 0.02 μg/kg/day TBEP (TBEP-L), (3) 0.2 μg/kg/day TBEP (TBEP-M), (4) 2 μg/kg/day TBEP (TBEP-H), (5) ovalbumin (OVA), (6) OVA + TBEP-L, (7) OVA + TBEP-M, and (8) OVA + TBEP-H (Figure 10). Tolerable daily intake of TBEP 2 μg/kg/day was set as high dose, and 1/10, 1/100 from high dose were set as medium and low doses. Each group contains 5 to 6 mice. TBEP was added to mouse chow for dietary exposure. To obtain the desired dose, mice were fed with a diet containing 0, 1.67, 16.7, or 167 μg TBEP per 10 kg of AIN-76A rodent chow (Japan Clea Co., Tokyo, Japan) from 5 to 11 weeks old. These doses were calculated based on the estimated consumption of 3 g of chow per mouse per day. TBEP-H dose is equivalent to the TDI defined by the Swiss Federal Office of Public Health [34]. The mice can take food and water freely. Mice were housed in an animal unit maintained at 22–26 °C and 40–69% humidity with a 12 h–12 h light-dark cycle. Daily food intake was checked, and the mice were weighed biweekly. Once every two weeks, the mice were anesthetized with isoflurane (FUJIFILM Wako Pure Chemical Corporation, Osaka, Japan) and given intratracheal instillation of 50 μL phosphate-buffered saline (PBS; pH7.4; Thermo Fisher Scientific, Inc., Rockford, IL, USA) alone or 1 μg of OVA (20 μg/mL, Sigma-Aldrich Co., St Louis, MO, USA). At 11 weeks of age, a novel object recognition test was performed 2 h after the last instillation. Forty-eight hours after the last instillation, the mice were euthanized with an intraperitoneal injection of sodium pentobarbital (150 mg/kg) for brain sample collection. This experimental protocol was approved by the Ethics Committee of the Animal Care and Experimentation Council of the National Institute for Environmental Studies (NIES), Japan (AE-18-17, AE-17-10). All efforts were made to minimize the number of animals used and their suffering.

### 4.2. Novel Object Recognition Test

The novel object recognition test was conducted to assess the ability of mice to recognize a new object from a familiar one. We performed a novel object recognition test over a period of 4 days, including a habituation phase (15 min/day for two consecutive days), a training phase (10 min for 1 day), and a test phase (5 min for 1 day) in each mouse at 11-week age as previously [35]. During the habituation phase, the mouse was placed in a rectangular cage (50 × 50 × 40 cm) made of acryl for 15 min per day for 2 days without an object. Then, during the training phase, two identical objects (6 × 7 × 8 cm) were placed near the corners on one wall of the rectangular cage (10 cm from each adjacent wall). The mouse was placed into the center of the cage facing the opposite wall and was allowed to explore both objects for 10 min. Exploration was defined as the mouse pointing its nose toward the object at a distance of less than 2 cm. We did not record the time spent sitting or resting against the object. Twenty-four hours after the training phase, during the test phase, one of the old objects was replaced with a novel object (8 × 9 × 10 cm) and was presented to each mouse for 5 min. To control for odor cues, the open field arena and the objects were thoroughly cleaned with water, dried, and ventilated for a few minutes between mice. The object exploration time was recorded using a video-assisted tracking system (Muromachi Kikai Co. Ltd., Tokyo, Japan). Discrimination between two objects was calculated using a discrimination index (DI) as follows: DI = ([novel object exploration time/total exploration time] − [familial object exploration time/total exploration time]) × 100. This equation takes into account individual differences in the total amount of exploration time [36]. The positions of the objects in the test and the objects used as a novel or familiar were counterbalanced between the mice.

### 4.3. Quantification of mRNA Expression Levels

At 11 weeks of age, after novel object recognition test, the mice (n = 5–6 from each group) were sacrificed under deep pentobarbital anesthesia, and the hippocampus was collected from each group of mice and frozen quickly in liquid nitrogen, then stored at –80 °C until the extraction of the total RNA. Briefly, the total RNA was extracted from the hippocampal samples using the BioRobot EZ-1 and EZ-1 RNA tissue mini kits (Qiagen GmbH, Hilden, Germany). Then, the purity of the total RNA was examined, and the quantity was estimated using the ND-1000 NanoDrop RNA Assay protocol (NanoDrop, Wilmington, DE, USA), as described previously [35]. Next, we performed first-strand cDNA synthesis from the total RNA using SuperScript RNase H-Reverse Transcriptase II (Invitrogen, Carlsbad, CA, USA), according to the manufacturer’s protocol. We examined the hippocampal mRNA expression levels using a quantitative real-time RT-PCR method (Light Cycler 96, Roche, Germany). The tissue 18S rRNA level was used as an internal control. The primer sequences used in the present study are shown below. Some primers 387 (*Nr1*, NM_008169; *Nr2a*, NM_008170; *Nr2b*, NM_008171, *Il-1β*, NM_008361; *Cox2*, NM_011198; *Ho1*, NM_010442; *Iba1*, NM_019467) were purchased from Qiagen, Sample and Assay Technologies. Other primers were designed in our laboratory as follows: *18S* (forward 5′-TACCACATCCAAAAGGCAG-3′, reverse 5′-TGCCCTCCAATGGATCCTC-3′), *Tnf-α* (forward 5′-GGTTCCTTTGTGGCACTTG-3′, reverse 5′-TTCTCTTGGTGACCGGGAG-3′) and *Gfap* (forward 5′-AGAAGCTCCAGGATGAAACC-3, reverse 5′-AGCGACTCAATCTTCCTCTC-3′). Data were analyzed using the comparative threshold cycle method. Then, the relative mRNA expression levels were expressed as mRNA signals per unit of *18S* rRNA expression.

### 4.4. Immunohistochemical Analyses

Microglia are major immune cells of the brain, and microglial activation indicates neurotoxicity. To detect microglial activation in the hippocampus, the hippocampal tissue sections were immunostained with microglial marker *Iba1* as described previously [37]. Briefly, the brain sections were immersed in absolute ethanol followed by 10% H_2_O_2_ for 10 min each at room temperature. After rinsing in 0.01-M phosphate buffer saline, the sections were blocked with 2% normal swine serum in PBS for 30 min at room temperature and then reacted with goat polyclonal anti-Iba1 (diluted 1:100; abcam: ab5076; Tokyo, Japan) in PBS for 1 h at 37 °C. Then, the sections were reacted with biotinylated donkey anti-rabbit IgG (1:300 Histofine; Nichirei Bioscience, Tokyo, Japan) in PBS for 1 h at 37 °C. The sections were then incubated with peroxidase-tagged streptavidin (1:300, ABC KIT) containing PBS for 1 h at room temperature. After a further rinsed in PBS, Iba1 immunoreactivity was detected using a Dako DAB Plus Liquid System (Dako Corp., Carpinteria, CA, USA). To detect the immunoreactivity of Iba1 in the hippocampus, photomicrographic digital images (150 dpi, 256 scales) of the hippocampal regions were taken using a CCD camera connected to a light microscope. Numbers of Iba1-positive microglia located within the DG area of the hippocampus were quantified in high power field using ImageJ software (5 fields per section and 3 sections per mouse, n = 3 mice/group).

### 4.5. Mast Cell Staining

Mast cells are found in the connective tissue, and their cytoplasm contains granules composed of acidic compounds such as heparin and histamine. To detect mast cells in the hippocampus, the hippocampal tissue sections were rinsed in two changes of distilled water. Then, the sections were incubated in toluidine blue solution (0.5% in 60% ethanol, pH 2.0) (Sigma-Aldrich, St. Louis MO, USA) (n = 3/group) for 5 min. After that, it was rinsed in three changes of distilled water, dehydrated, and mounted with synthetic resin. Mast cells were detected in these sections by red-purple and the background blue.

### 4.6. Immunostaining of Brain Sections with Fc Epsilon R1α (FcεR1α)

Mice were given flush-perfusion with phosphate-buffered saline (PBS) followed by perfusion-fixation with 4% paraformaldehyde (PFA). The removed brain was immersed in 4% paraformaldehyde and subsequently in 30% sucrose for 48 h at 4 °C. After the brain had been cut into coronal 30-μm sections, H_2_O_2_-inactivation of endogenous peroxidase activity was performed for immunohistochemistry, followed by treatment with protein block serum-free ready-to-use (Dako, Santa Clara, CA, USA) for 1 h at room temperature to block non-specific protein binding. The sections were incubated with a mouse monoclonal antibody against *FcεRIα* (9E1, ab54411, abcam^®^, Cambridge, UK) in antibody diluent with background reducing components (Dako, Santa Clara, CA, USA) for overnight at 4 °C. After incubation with an HRP-conjugated secondary antibody, the sections were then treated with 3,3′-diaminobenzidine (DAB) for immunohistochemistry.

### 4.7. Statistical Analysis

All the data were expressed as the mean ± standard error (S.E.). The statistical analysis was performed using the StatMate II statistical analysis system for Microsoft Excel, Version 5.0 (Nankodo Inc., Tokyo, Japan). The data were analyzed using a one-way analysis of variance with a post-hoc analysis using the Bonferroni/Dunn method. Differences were considered significant at *p* < 0.05.

## 5. Conclusions

The summarized results of this study are shown in Figure 11. Oral exposure to TBEP induced neuroinflammation via upregulation of proinflammatory cytokines and oxidative stress markers in mice with or without OVA immunization and could impair memory function accompanied with abnormal activation of *Nmda* receptor subunits. Increased expression of mast cells, microglia, and astrocytes was observed in an allergic asthmatic mouse model. Neuroimmune crosstalk between neurons, glia cells, and mast cells and their secretory molecules might play a key role in response to environmental chemical exposure. Further studies are needed to elucidate the role of brain immune cells in environmental chemical-induced neurobehavioral toxicity.

## Figures and Tables

**Figure 1 ijms-23-00655-f001:**
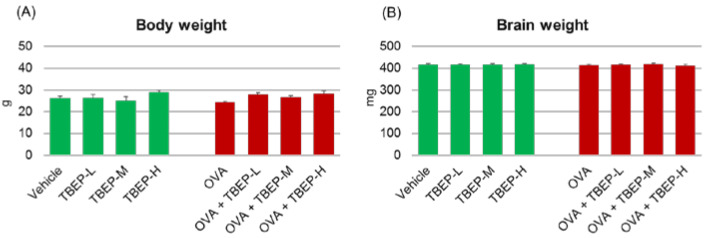
Assessment of general toxicity. (**A**) Body weight and (**B**) brain weight of male mice exposed to TBEP with or without OVA immunization.

**Figure 2 ijms-23-00655-f002:**
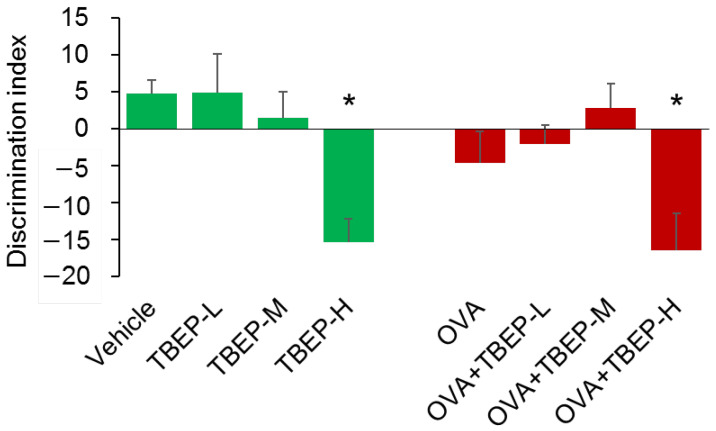
Novel object recognition test. Discrimination index between familial and novel objects was measured (n = 6, * *p* < 0.05 vs. vehicle group).

**Figure 3 ijms-23-00655-f003:**
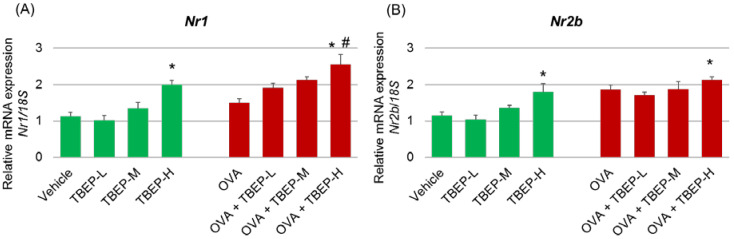
Messenger RNA expression levels of memory function-related genes *Nmda* receptor subunits. (**A**) *Nr1* and (**B**) *Nr2b* mRNA in the hippocampus of TBEP-exposed male mice with or without OVA immunization. (n = 6, * *p* < 0.05 vs. vehicle alone; # *p* < 0.05 vs. OVA alone).

**Figure 4 ijms-23-00655-f004:**
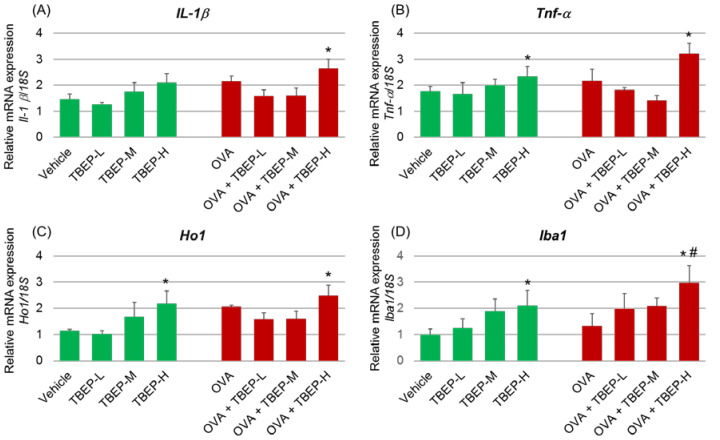
Messenger RNA expression levels of inflammatory molecular markers. (**A**) *Il-1β*, (**B**) *Tnf-α*, (**C**) *Ho1* and (**D**) *Iba1* mRNA in the hippocampus of TBEP-exposed male mice with or without OVA immunization. (n = 6, * *p* < 0.05 vs. vehicle alone; # *p* < 0.05 vs. OVA alone).

**Figure 5 ijms-23-00655-f005:**
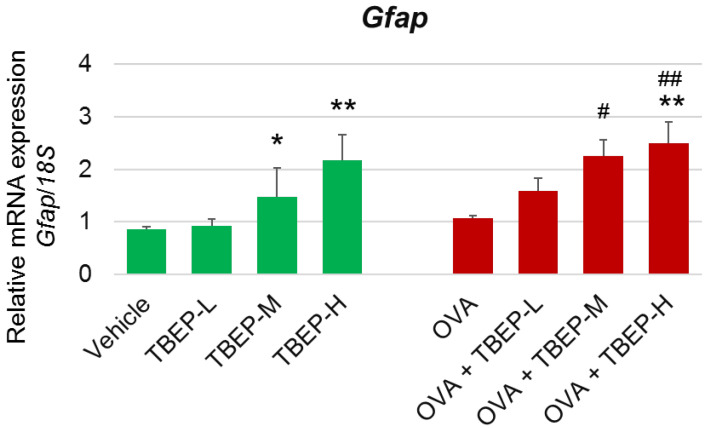
Messenger RNA expression levels of astrocyte marker in the hippocampus. *Gfap* mRNA in the hippocampus of TBEP-exposed male mice with or without OVA immunization. (n = 6, * *p* < 0.05, ** *p* <0.01 vs. vehicle alone; # *p* < 0.05, ## *p* < 0.01 vs. OVA alone).

**Figure 6 ijms-23-00655-f006:**
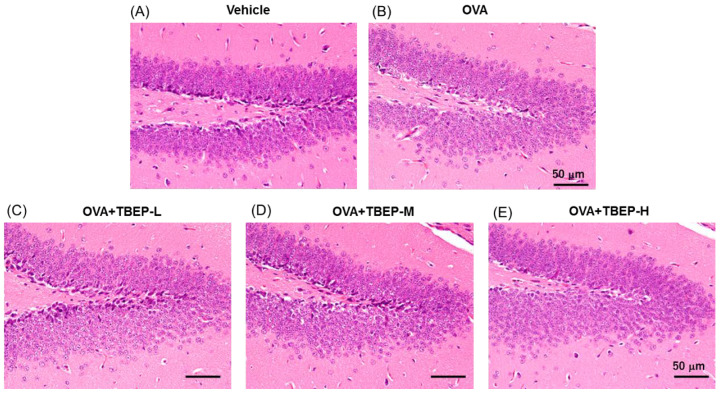
Histological analysis of dentate gyrus of the hippocampus. H&E stain in (**A**) vehicle, (**B**) OVA, (**C**) OVA + TBEP-L, (**D**) OVA + TBEP-M and (**E**) OVA + TBEP-H. (Scale bar = 50 μm).

**Figure 7 ijms-23-00655-f007:**
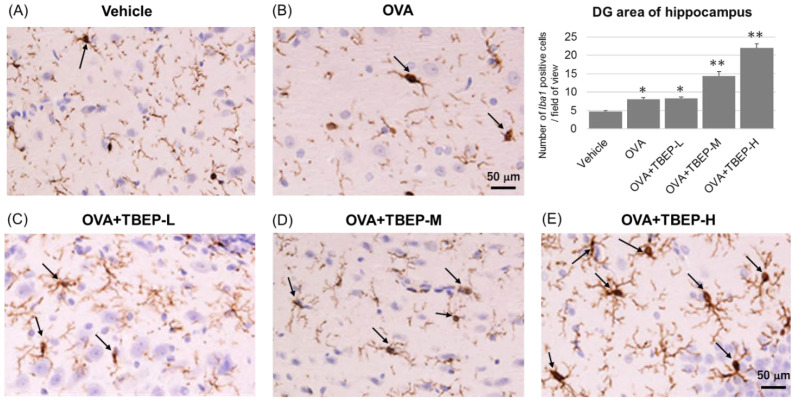
Representative photomicrographs of the dentate gyrus of the hippocampus. Microglia marker *Iba1* immunoreactivity in (**A**) vehicle, (**B**) OVA, (**C**) OVA + TBEP-L, (**D**) OVA + TBEP-M and (**E**) OVA + TBEP-H. Black arrows indicate activated microglia. (Scale bar = 50 μm). Number of *Iba1*-positive cells located within the DG area of the hippocampus was quantified in a high power field under microscopy (histogram). (n = 3, * *p* < 0.05, ** *p* < 0.01 vs. vehicle alone).

**Figure 8 ijms-23-00655-f008:**
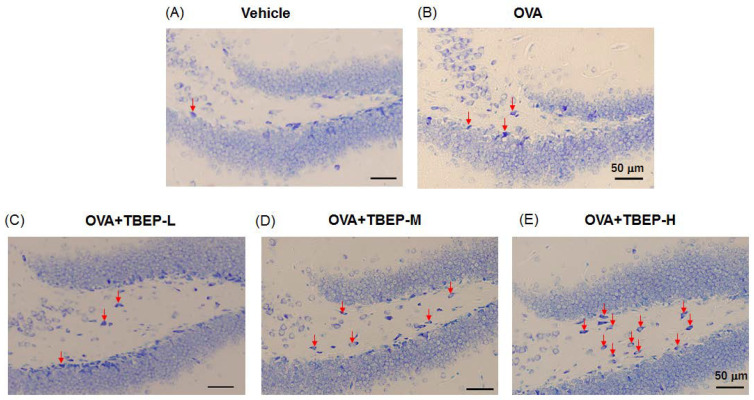
Toluidine blue staining of the dentate gyrus of the hippocampus. Toluidine blue stain showing mast cells in (**A**) vehicle, (**B**) OVA, (**C**) OVA + TBEP-L, (**D**) OVA + TBEP-M, and (**E**) OVA + TBEP-H. Red arrows indicate mast cells. (Scale bar = 50 μm).

**Figure 9 ijms-23-00655-f009:**
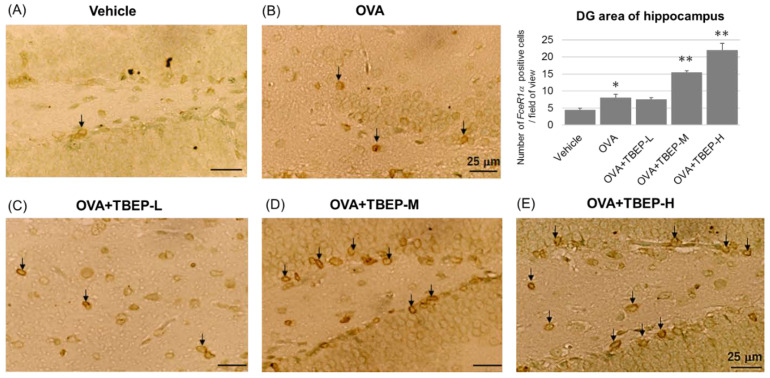
Immunohistochemical analysis of *FcεR1α* (mast cell marker) of the dentate gyrus of the hippocampus. *FcεR1α*-positive mast cells in (**A**) vehicle, (**B**) OVA, (**C**) OVA + TBEP-L, (**D**) OVA + TBEP-M and (**E**) OVA + TBEP-H. Black arrows indicate *FcεR1α*-positive mast cells. (Scale bar = 25 μm). Number of *FcεR1α*-positive cells located within the DG area of the hippocampus was quantified in a high power field under microscopy (histogram). (n = 2, * *p* < 0.05, ** *p* < 0.01 vs. vehicle alone).

**Figure 10 ijms-23-00655-f010:**
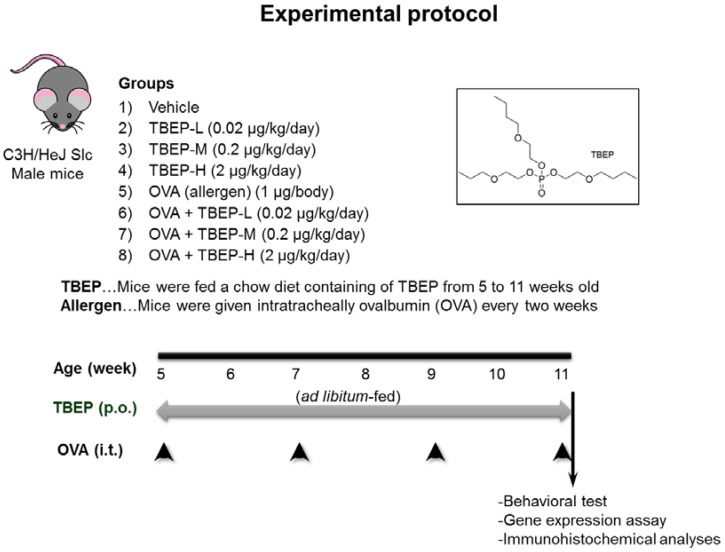
Experimental protocol. Five-week-old mice were allotted into eight groups as follows: (1) vehicle, (2) 0.02 μg/kg/day TBEP (TBEP-L), (3) 0.2 μg/kg/day TBEP (TBEP-M), (4) 2 μg/kg/day TBEP (TBEP-H), (5) ovalbumin (OVA), (6) OVA + TBEP-L, (7) OVA + TBEP-M, and (8) OVA +T BEP-H. Each group contains 5 to 6 mice. TBEP was added to mouse chow for dietary exposure. Once every two weeks, the mice were given intratracheal instillation of 1 μg of OVA. At 11 weeks of age, a novel object recognition test was performed. Then, the hippocampi were collected to detect neurological and immunological molecular markers.

**Figure 11 ijms-23-00655-f011:**
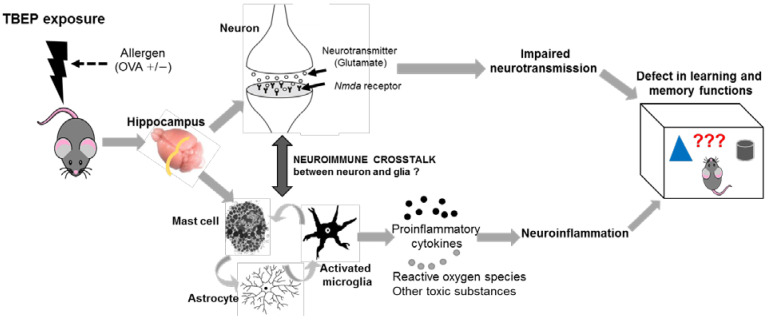
Diagram of summarized results of TBEP-induced neurotoxicity. Dietary exposure to TBEP could impair memory function accompanied by abnormal activation of *Nmda* receptor subunits and induced neuroinflammation via upregulation of proinflammatory cytokines and oxidative stress markers in mice with or without OVA immunization. Increased expression of mast cells and glial cells (microglia and astrocytes) activation was observed in an allergic asthmatic mouse model. Neuroimmune crosstalk between neurons, immune cells such as mast cells, and glial cells (microglia and astrocyte) inflammatory molecules might play a key role in response to environmental chemical-induced neurobehavioral toxicity.

## Data Availability

The data presented in this study are available on request from the corresponding author.

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
