# Peer review of "Dietary Exposure to Flame Retardant Tris (2-Butoxyethyl) Phosphate Altered Neurobehavior and Neuroinflammatory Responses in a Mouse Model of Allergic Asthma"

_ijms, 2022, doi:10.3390/ijms23020655_

Round 1
Reviewer 1 Report
Win-Shwe et al. provides here a very interesting work with compelling data regrading TBEP and its danger. The animal work is well-designed with an appropriate number of animals for a pilot study. The authors are doing an outstanding work describing the experimental protocol. The figure 8 is clear and helpful in this regard. While the animal study is well designed the histology studies must be improved or complemented by additional biochemical or molecular analyses as well as quantitative analysis without these this interesting work seems really impaired.
Minor comments:
there is no alpha to TNF, also when refereeing to mouse gene or mRNA the gene should be italicized and only the first letter is capitalized. The nomenclature should be correct and consistent across the manuscript.
Major comments:
Figure 6: the microglia staining is convincing however a density per field of view would significantly improve the quality of the analysis and provide quantitative data. The activated status of the microglia based on their morphology should also be evaluated from the images.
Toluidine blue staining can be used to stain mast cells, it however can also label neurons. Adding immunohistochemical analysis would be more convincing regarding the presence of mast cells. For exemple CD117 staining in combination with the toluidine blue could be used to convincedly label mast cells. Alternatively, transcript enriched in mast cells could be used to support the analysis. Following the staining a quantification must be provided.
In combination to the microgliosis and mast cell infiltration evaluating the astrogliosis would be a good addition to the present work.
Author Response
Responses to the Reviewer’ comments
Reviewer#1
Comments and Suggestions for Authors
Win-Shwe et al. provides here a very interesting work with compelling data regarding TBEP and its danger. The animal work is well-designed with an appropriate number of animals for a pilot study. The authors are doing an outstanding work describing the experimental protocol. The figure 8 is clear and helpful in this regard. While the animal study is well designed the histology studies must be improved or complemented by additional biochemical or molecular analyses as well as quantitative analysis without these this interesting work seems really impaired.
Responses: First of all, we would like to appreciate the Reviewer for invaluable comments and suggestions. As commented by the Reviewer, we added molecular and immunohistochemical analyses in our revised manuscript.
Minor comments:
There is no alpha to TNF, also when refereeing to mouse gene or mRNA the gene should be italicized and only the first letter is capitalized. The nomenclature should be correct and consistent across the manuscript.
Responses: As suggested by the Reviewer, we corrected the writing style of symbol of genes throughout the revised manuscript.
Major comments:
Figure 6: the microglia staining is convincing however a density per field of view would significantly improve the quality of the analysis and provide quantitative data. The activated status of the microglia based on their morphology should also be evaluated from the images.
Responses: As commented by the Reviewer, we added density and number of Iba1-positive activated microglia and morphology of activated microglia in Figure 7 in the revised manuscript.
Toluidine blue staining can be used to stain mast cells, it however can also label neurons. Adding immunohistochemical analysis would be more convincing regarding the presence of mast cells. For exemple CD117 staining in combination with the toluidine blue could be used to convincedly label mast cells. Alternatively, transcript enriched in mast cells could be used to support the analysis. Following the staining a quantification must be provided.
Responses: As suggested by the Reviewer, we examined the expression of mast cell surface marker Fc epsilon RIa (FcεR1α) and expressed immunostaining and number of FcεRIα-positive-mast cells in the hippocampus in Figure 9 in the revised manuscript.
In combination to the microgliosis and mast cell infiltration evaluating the astrogliosis would be a good addition to the present work.
Responses: As commented by the Reviewer, to detect the astrogliosis, we examined the astrocyte marker glial fibrillary acidic protein (Gfap) and found its mRNA expression was significantly increased in TBEP-M and TBEP-H groups with or without OVA compared with the vehicle alone or OVA alone group (Figure 5) in the revised manuscript.

Reviewer 2 Report
The study on the safety of a household fire retardant is interesting, but it is not clear to me why a model of allergic asthma was chosen to study the effect of this substance on neuroinflammation and behavioral responses. What is the link between allergic asthma and neuroinflammation as a result of TBEP exposure? Moreover, why was this method of administration chosen? If the substance is a fire retardant and floor care product, it is likely that it will enter the human body through the respiratory tract and not per os. The authors refer to an apparently irrelevant articles (published over 20 years ago) on the discovery of TBEP in surface and ground water. However, its detection in groundwater has nothing to do with its presence in drinking water. Is there any evidence that this substance is found in large quantities in drinking water, which undergoes rigorous quality control?
Thus, the relevance of this study, as well as the experimental model, is not substantiated. The meager formulations of the authors do not strike me as convincing.
The authors showed that under the action of a high dose of TBEP, some pro-inflammatory molecules are up-regulated. However, there is no evidence that this led to an actual increase in the amount of these pro-inflammatory cytokines in cells, which could be shown by Western blotting or ELISA.
Figure 9 is unreasonably located in section 4.4. of the Materials and Methods.
Author Response
Responses to the Reviewer 2’s comments
Reviewer#2
Comments and Suggestions for Authors
The study on the safety of a household fire retardant is interesting, but it is not clear to me why a model of allergic asthma was chosen to study the effect of this substance on neuroinflammation and behavioral responses. What is the link between allergic asthma and neuroinflammation as a result of TBEP exposure? Moreover, why was this method of administration chosen? If the substance is a fire retardant and floor care product, it is likely that it will enter the human body through the respiratory tract and not per os. The authors refer to an apparently irrelevant articles (published over 20 years ago) on the discovery of TBEP in surface and ground water. However, its detection in groundwater has nothing to do with its presence in drinking water. Is there any evidence that this substance is found in large quantities in drinking water, which undergoes rigorous quality control?
Thus, the relevance of this study, as well as the experimental model, is not substantiated. The meager formulations of the authors do not strike me as convincing.
Responses: We would like to apologize for missing clear and precise explanation for selection of an allergic asthmatic mouse model in our present study. The reason to choose an allergic asthmatic model was mentioned in “Introduction session” in our revised manuscript as follow:
Previously, we have described possible brain–lung network in asthma (Win-Shwe et al., 2019). Briefly, asthma may induce brain hypoxia and cause inflammatory responses via neuro-immune molecular markers, and which induce cognitive impairment via hippocampal memory function related genes. We have reported that exposure to environmental chemicals enhanced these effects (Win-Shwe et al., 2007; 2013; 2019). Recent study has also demonstrated that asthma-induced brain hypoxia caused the impairment of learning and memory via dysfunction of synaptic tissues and blood vessels, increased levels of hypoxia-inducible factors marker HIF-1α and HIF-2α (Ren et al, 2021). Therefore, in this study, we hypothesized that chemical alone, asthma alone or co-exposure of chemical and allergen induces brain hypoxia causes memory impairment via modulation of hippocampal synaptic genes and inflammatory mediators released from brain immune and glia cells.
Regarding method of administration, we selected dietary exposure method in the present study. Flame retardants are commonly found in indoor air, house dust and hand wipes, thus, inhalation, hand-to-mouth and dermal contact are all important routes of human exposure to FRs. Moreover, migration of plasticizer from food packaging materials and drinking water bottle may consider for dietary or diet-associated intake of OPFR. Thus, as commented by the Reviewer, inhalation is major route of entry to human body, however, dietary route is another route of entry of FRs.
As commented by the Reviewer, we deleted irrelevant articles (published over 20 years ago) on the discovery of TBEP in surface and ground water in our revised manuscript. We added relevant and recent references as follows in our revised manuscript “It has been reported that daily exposure to OPFRs via indoor dust [He et al., 2018] and diet and diet-associated intake are higher [He et al., 2018]. Thus, it was suggested that common routes of entry of TBEP to body are inhalation of indoor dust or ingestion of contaminated food.”
The authors showed that under the action of a high dose of TBEP, some pro-inflammatory molecules are up-regulated. However, there is no evidence that this led to an actual increase in the amount of these pro-inflammatory cytokines in cells, which could be shown by Western blotting or ELISA.
Responses: As suggested by the Reviewer, it should be checked up-regulation of proinflammatory cytokines by Western blotting or ELISA method to detect protein levels. However, we have collected hippocampal samples for mRNA analysis and whole brain samples for histological and immunohistochemical analyses. We have a plan to detect protein levels of molecular markers using Western blotting or ELISA in our future study.
Figure 9 is unreasonably located in section 4.4. of the Materials and Methods.
Responses: As commented by the Reviewer, our summarized Figure 11 (in revised manuscript) was moved to Discussion section, under Conclusion.

Round 2
Reviewer 1 Report
Dear authors,
thank you for addressing some of my comments, the figure 9 data provide stronger support than the Toluidine blue staining. The staining of the mast cells is now convincing and a significant addition to the manuscript.
However:
the alpha following Tnf has not been corrected in the figures.
the weight of the animal in figure 1 must be wrong: ~400g/mouse
why some genes are still labelled using the human nomenclature while the other are correctly named using the mouse nomenclature.
in figure 7 the axis of the graph should read number of IBA1 positive cells per field of view or a given area.
the corresponding modification should be performed in the figure 9.
Author Response
Response to the Reviewer #1’s comments
Dear authors,
thank you for addressing some of my comments, the figure 9 data provide stronger support than the Toluidine blue staining. The staining of the mast cells is now convincing and a significant addition to the manuscript.
However:
the alpha following Tnf has not been corrected in the figures.
Response: We appreciate for pointing out our missing to correct. We corrected Tnf-a in Figure 4B in our revised manuscript (R2).
the weight of the animal in figure 1 must be wrong: ~400g/mouse
Response: We would like to thank the Reviewer’s for detail checking. We corrected “Body weight and brain weight” in Figure 1 in our revised manuscript (R2).
why some genes are still labelled using the human nomenclature while the other are correctly named using the mouse nomenclature.
Response: As commented by the Reviewer, we corrected all genes in text and figures in our revised manuscript (R2).
in figure 7 the axis of the graph should read number of IBA1 positive cells per field of view or a given area.
Response: As suggested by the Reviewer, we added Iba1 positive cells per field of view in Figure 7 in our revised manuscript (R2).
the corresponding modification should be performed in the figure 9.
Response: As suggested by the Reviewer, we added FcεRIα positive cells per field of view in Figure 9 in our revised manuscript (R2).

Reviewer 2 Report
The authors made the necessary changes to the manuscript. I believe that the manuscript can be published in this form.
Author Response
Response to the Reviewer #2’s comments
The authors made the necessary changes to the manuscript. I believe that the manuscript can be published in this form.
Response: We would like to thank the Reviewer #2 for invaluable comments and advise for improvement of our manuscript.
